# Plasma Lipid Profiles Change with Increasing Numbers of Mild Traumatic Brain Injuries in Rats

**DOI:** 10.3390/metabo12040322

**Published:** 2022-04-02

**Authors:** Chidozie C. Anyaegbu, Harrison Szemray, Sarah C. Hellewell, Nathan G. Lawler, Kerry Leggett, Carole Bartlett, Brittney Lins, Terence McGonigle, Melissa Papini, Ryan S. Anderton, Luke Whiley, Melinda Fitzgerald

**Affiliations:** 1Curtin Health Innovation Research Institute, Curtin University, Bentley, WA 6102, Australia; chidozie.anyaegbu@curtin.edu.au (C.C.A.); carole.bartlett@curtin.edu.au (C.B.); brittney.lins@curtin.edu.au (B.L.); terence.mcgonigle@curtin.edu.au (T.M.); melissa.papini@curtin.edu.au (M.P.); 2Perron Institute for Neurological and Translational Science, Ralph and Patricia Sarich Neuroscience Research Institute Building, Nedlands, WA 6009, Australia; kerry.leggett@uwa.edu.au (K.L.); ryan.anderton@nd.edu.au (R.S.A.); 3Australian National Phenome Centre, Health Futures Institute, Murdoch University, Murdoch, WA 6150, Australia; 34255866@student.murdoch.edu.au (H.S.); nathan.lawler@murdoch.edu.au (N.G.L.); 4Centre for Computational and Systems Medicine, Health Futures Institute, Murdoch University, Murdoch, WA 6150, Australia; 5School of Health Sciences, The University of Notre Dame Australia, Fremantle, WA 6160, Australia

**Keywords:** lipidomics, lipid, mild traumatic brain injury, repeated mild traumatic brain injury, liquid chromatography–mass spectrometry

## Abstract

Mild traumatic brain injury (mTBI) causes structural, cellular and biochemical alterations which are difficult to detect in the brain and may persist chronically following single or repeated injury. Lipids are abundant in the brain and readily cross the blood-brain barrier, suggesting that lipidomic analysis of blood samples may provide valuable insight into the neuropathological state. This study used liquid chromatography-mass spectrometry (LC-MS) to examine plasma lipid concentrations at 11 days following sham (no injury), one (1×) or two (2×) mTBI in rats. Eighteen lipid species were identified that distinguished between sham, 1× and 2× mTBI. Three distinct patterns were found: (1) lipids that were altered significantly in concentration after either 1× or 2× F mTBI: cholesterol ester CE (14:0) (increased), phosphoserine PS (14:0/18:2) and hexosylceramide HCER (d18:0/26:0) (decreased), phosphoinositol PI(16:0/18:2) (increased with 1×, decreased with 2× mTBI); (2) lipids that were altered in response to 1× mTBI only: free fatty acid FFA (18:3 and 20:3) (increased); (3) lipids that were altered in response to 2× mTBI only: HCER (22:0), phosphoethanolamine PE (P-18:1/20:4 and P-18:0/20:1) (increased), lysophosphatidylethanolamine LPE (20:1), phosphocholine PC (20:0/22:4), PI (18:1/18:2 and 20:0/18:2) (decreased). These findings suggest that increasing numbers of mTBI induce a range of changes dependent upon the lipid species, which likely reflect a balance of damage and reparative responses.

## 1. Introduction

Mild traumatic brain injury (mTBI) is increasingly recognised as a substantial public health problem, particularly in the sporting context, where repeated mTBI (rmTBI) is prevalent. At least 20% of the 3.2 million people worldwide that present to hospital each year experience persistent neuropsychological, cognitive, sleep or physical dysfunction [1], with those experiencing rmTBI at greater risk [2,3]. rmTBI is also associated with an increased risk of developing neurodegenerative diseases such as dementia and Parkinson’s disease [4,5]. There are no effective treatments for the pathobiology perpetuating mTBI symptoms, partly due to an incomplete understanding of the underlying mechanisms of the condition. Longitudinal mechanistic studies are challenging due to the small number of accessible indicators of damage that reflect pathology in the brain following single and repeated mTBI.

A rapid-onset pathophysiological feature of mTBI is terminal membrane depolarisation caused by the excessive release of neurotransmitters from damaged neurons and glia. This activates voltage-gated calcium and sodium channels, leading to cell membrane and protein degradation by lipases and proteases [6]. The associated increase in intracellular calcium and sodium levels also triggers excessive reactive oxygen species production, in turn causing oxidative damage to proteins, lipids, and other macromolecules [7]. Such pathophysiological changes can cause degradation of the axonal cytoskeleton and extracellular matrix, resulting in the release of cellular components usually absent in the blood [8].

There is growing evidence from experimental and clinical studies that TBI may induce unique alterations in brain lipid species which can be detected in the blood [9,10]. These studies support the concept that changes to lipids and their metabolites in the blood may reflect disruptions to the structural integrity of lipid-rich cell membranes in the brain. Depending on their structure and function, lipids can be classified into two categories: (1) lipids that lack fatty acids (e.g., cholesterol and vitamins) and (2) fatty acid-containing lipids [9]. The latter class includes storage lipids (e.g., mono-, di-, and triacylglycerols) and membrane lipids such as glycerophospholipids (e.g., phosphatidylcholine, PC; phosphatidylethanolamine, PE; phosphatidylserine, PS; and phosphatidylinositol, PI) and sphingolipids (e.g., ceramide, sphingomyelin (SM) and glycosphingolipids). Membrane phospholipid metabolites include lysophosphatidylcholine (LPC), lysophosphatidylethanolamine (LPE), lysophosphatidylglycerol (LPG) and lysophosphatidylinositol (LPI). Lipids account for 60% of the brain’s dry weight, and ~75% of lipids in mammals are specific to neural tissues, highlighting the unique importance of lipids for brain function [11]. Indeed, lipids play several key roles in the central nervous system, including cell membrane formation, regulation of membrane-bound ion channels, neurotransmission, and immune signalling [12]. Essential fatty acids for generating cell membrane phospholipids are either synthesised in the brain or transported from the systemic circulation through the blood–brain barrier [13,14].

Plasma lipid levels are altered in both human and rodent mTBI studies, with predominant focus being on the acute (<3 days) and chronic (>1 month) stages of injury. Clinical studies of populations vulnerable to rmTBI, such as contact sports players and military personnel, have reported decreases in plasma markers both acutely and chronically, with a cohort of ice hockey players demonstrating decreases in plasma PC and LPC 55 hours post-injury [15]. In soldiers who had experienced mTBI up to 10 years prior to lipid analysis, significant decreases in plasma levels of PC, LPC, PE, LPE, PI and SM were also detected and, interestingly, these changes were more pronounced in soldiers who had developed post-traumatic stress disorder, suggesting that lipid alterations may be more pronounced as pathology increases [16]. Similarly, in a closed-head mouse model of mTBI, the patterns of chronic decrease in plasma PC, LPC, PE, LPE and PI observed at 3, 12 and 24 months post-injury mimicked those detected in the soldiers with chronic mTBI [10]. In contrast to these peripheral decreases in plasma phospholipids, significant increases in PE, LPE, PC, LPC and SM have been reported in the mouse brain at acute (24 h) and/or chronic (3, 6, 9 and 12 month) time points post-rmTBI [17], suggesting that concomitant decreases and increases in plasma and brain lipids, respectively, may reflect lipid transport to the brain as a reparative response. However, the changes in plasma phospholipids at the subacute phase of rmTBI (day 5–14) have not been explored in a closed-head rodent model of rmTBI that incorporates rotational movements, the predominant mechanism of tissue shearing and traumatic unconsciousness in human mTBI [18]. A well-characterised, rotational acceleration rat model of closed-head mTBI [19,20,21,22], allows the generation of cellular lipid alterations likely to be of clinical relevance. In this study, we used liquid chromatography–mass spectrometry (LC-MS) in conjunction with our rat rotational model of closed head mTBI [19,20,21,22] to profile the plasma lipidome at day 11 following sham, one (1×) or two (2×) mTBI, hypothesising that changes would be observed with increasing numbers of mTBI relative to sham injury. Analyses were performed at day 11 because our previous studies showed that the cellular indicators of damage, including microglial reactivity and oxidative stress, were most apparent at this subacute time [19,20], as compared to the more acute (day 4; [21]) and chronic (3 month; [22]) time points. Female rats were used to enable comparison of findings to our previous work [19,20,21,22], and to address the substantial under-representation of female animals in TBI literature.

## 2. Results

### 2.1. Orthogonal Projection to Latent Structure-Discriminant Analysis (OPLS-DA) of Plasma Lipids Discriminates Sham, 1× mTBI and 2× mTBI

Following data cleaning, the final dataset consisted of 856 reproducible lipid species from 19 different lipid classes. A principal component analysis (PCA) model was created to assess the overall variation in the dataset (Appendix A). From the PCA, the replicate quality control (QC) samples successfully clustered, demonstrating the reproducibility of the extraction and analysis.

To identify the metabolic signature associated with mTBI, supervised orthogonal projection to latent structure-discriminant analysis (OPLS-DA) was performed on the lipid data generated from the sham (*n* = 4), 1× mTBI (*n* = 6) and 2× mTBI (*n* = 7) groups (Figure 1). The OPLS-DA model scores were as follows: sham control and 1× mTBI groups (Figure 1A and Appendix A) (R2 X = 0.637, R2 Y = 0.858, Q2 = 0.307); 1× mTBI and 2× mTBI groups (Figure 1B and Appendix A) (R2 X = 0.558, R2 Y = 0.84, Q2 = 0.206); sham control and 2× mTBI groups (Figure 1C and Appendix A) (R2 X = 0.603, R2 Y = 0.952, Q2 = 0.322).

Finally, to test for the existence of a progressive trend in the data following the 1× mTBI and 2× mTBI interventions, the 1× mTBI data were predictively projected onto the existing OPLS-DA model for sham control vs. 2× mTBI. This resulted in the 1× mTBI data projecting into the model space between the sham control and 2× TBI groups, indicating a progressive change in the lipid data: sham control → 1× mTBI → 2× mTBI (Figure 1D).

### 2.2. Individual Plasma Metabolites Are Altered Differentially in Response to 1× or 2× mTBI

Given the results in the OPLS-DA models and to further stratify the three groups, a non-parametric Kruskal–Wallis one-way analysis of variance (ANOVA) on ranks was performed using the individual lipid species in the data set to further clarify which metabolites were driving the OPLS-DA models. This was performed on 856 individual lipid species identified in the data set, and a post hoc Dunn’s test was then completed to report inter-class differences. A total of 18 plasma lipids were found to be significantly different (*p* ≤ 0.05) between the sham control, 1× mTBI and 2× mTBI groups, respectively (Table 1).

These lipids could be grouped into three distinct patterns of change following injury. The first pattern was lipids whose concentration was significantly altered by both 1× mTBI or 2× mTBI (Figure 2): this included CE (14:0), which increased after 1× (*p* = 0.04) and 2× injuries (*p* < 0.01); PS (14:0/18:2) and HCER (d18:0/26:0), which decreased after 1× and 2× mTBI (PS (14:0/18:2): *p* < 0.01 for 1× and 2× mTBI; HCER (d18:0/26:0): *p* = 0.03 and *p* < 0.001 for 1× and 2× mTBI, respectively); and PI (16:0/18:2), which increased significantly after 1× mTBI (*p* < 0.01) but decreased in response to 2× mTBI (*p* = 0.01).

The second pattern of change was of lipids whose concentration was altered only in response to 1× mTBI (Figure 3). This pattern was specific to FFA, with significant increases in concentration of both FFA (18:3) (*p* = 0.05) and FFA (20:3) (*p* < 0.01) with 1× injury only. The third pattern was of lipids altered only in response to 2× mTBI (Figure 4). This pattern comprised an increase in the concentration of HCER (22:0) (*p* = 0.04) and PEs (P-18:1/20:4) (*p* = 0.02) and (P-18:0/20:1) (*p* = 0.05), and decreased concentrations of LPE (20:1) (*p* = 0.01), PC (20:0/22:4) (*p* = 0.05) and PIs (18:1/18:2) (*p* = 0.04) and (20:0/18:2) (*p* = 0.05), after 2× mTBI. Some lipid alteration was also observed between 1× and 2× mTBI (Appendix A). However, since there were no observed differences in comparison to the sham control, these findings were not considered in the scope of our hypotheses and current analyses.

## 3. Discussion

The metabolic changes associated with mTBI are complex and can have central and peripheral physiological effects. LC-MS can provide unique insight into chemical perturbations which occur following TBI [23,24,25]. Lipids are broadly known to be disturbed following TBI. However, the characterisation of alterations in individual lipid species after mTBI is an emerging line of enquiry [10,26,27]. Here, we applied a target lipid screen to measure 856 lipid species to investigate the effect of single vs. repeated mTBI on the plasma lipidome in a rat mTBI model at 11 days post-injury, a time at which the oxidative and inflammatory mediators of secondary pathology are evident [21,22]. We identified 18 significantly altered lipids in rat plasma that could accurately distinguish between sham, 1× mTBI and 2× mTBI, indicating that the lipidome remains significantly altered in the subacute phase of mTBI, and that this pathology increases variably with repeated injury.

Multivariate OPLS-DA (sham vs. 1× mTBI; sham vs. 2× mTBI; 1× mTBI vs. 2× mTBI) resulted in models with distinct separations between the classes. Furthermore, projection of the 1× mTBI intervention group onto the sham vs. 2× TBI model resulted in the 1× mTBI group being predicted centrally between the sham group and the 2× mTBI group, indicating that overall, lipid alterations follow progressive trends with increasing mTBI insults. These findings may provide further insight into the pathological consequences of increasing numbers of mTBI, as seen in contact sport and military injuries. Following multivariate modelling of the data, univariate analysis indicated eighteen lipids that distinguished between the sham, 1× mTBI and 2× mTBI groups. Further analysis of concentration ratios demonstrated three principal patterns of change across the 18 lipid biomarkers identified: (1) altered lipid concentration with both 1× or 2× mTBI vs. sham control; (2) altered lipid concentration with 1× mTBI (but not 2× mTBI); (3) lipid concentrations that significantly differed from sham control concentrations only after 2× mTBI (See Appendix A).

Cholesterol esters (CE) conformed to the first pattern, being increased after both 1× and 2× mTBI. CEs have been shown to increase transiently in a controlled cortical impact rat model of moderate/severe TBI, with CE (18:1) peaking at 3 days in the lesion and peri-lesional tissue [28]. However, altered CE (14:0) has not previously been reported in either a closed-head or an mTBI model. Our results suggest that the time course of CE alteration may be substantially different in closed-head injury, with significant elevations in plasma still detected at 11 days post-injury in our closed-head rotational mTBI model after 1× and 2× injuries. Indeed, the subacute elevations of lipids following this identified pattern warrant further investigation for their capacity as injury biomarkers, given that they were readily detectable in plasma beyond the acute phase. In contrast, we found that phosphoserine (PS) (14:0/18:2) and hexosylceramide (HCER) (d18:0/26:0) significantly decreased with both 1× and 2× mTBI. As an ester, PS is an important intermediary for phosphoric acid and serine. L-serine has received increasing attention as a multifunctional agent capable of restoring cognitive function, reducing inflammation and promoting remyelination in several neurological conditions [29], suggesting that downstream metabolite loss may reduce the production of this neurotrophic factor, with deleterious consequences. Lipid species from the phosphoinositol (PI) class (16:0/18:2) were increased in concentration in the 1× mTBI group when compared to the sham control group. However, the same species decreased in the 2× mTBI group. This finding is paradoxical, although it mirrors our previous findings of microglial reactivity and oxidative damage in brain tissue in this model [21,22], lending support to its biological plausibility. While these opposing responses to single and repeated injuries are currently under investigation, they reveal that pathology may not follow a linear pattern with increasing injury, but may instead involve distinct responses.

The second observed lipid pattern comprised free fatty acids (FFA), which were also increased only after 1× mTBI. FFAs are precursors to eicosanoids, which are important modulators of inflammation, cell signalling and vascular response (among others) throughout the central nervous system [30]. These products of lipid peroxidation have been increasingly reported to accumulate after TBI in both tissue and biofluids, potentiating the pathological response [25,31,32]. As such, FFA concentrations have been suggested as a putative clinical biomarker of outcome [32]. In the present study, our finding of increased plasma concentrations of both FFA (18:3) and FFA (20:3) with 1× mTBI supports these clinical findings, potentially demonstrative of persisting lipid peroxidation and inflammation in this subacute post-injury period. It is noteworthy that this pattern was not observed after 2× mTBI, and further investigation is required in a larger cohort to determine whether this FFA alteration is truly specific to single injury.

Our third observed pattern of lipids responsive only to 2× mTBI included HCER (22:0), PE (P-18:0/20:1), and PE (P-18:1/20:4), which increased with injury. These two PE species have been linked to multiple neurodegenerative diseases, including Parkinson’s disease and Alzheimer’s disease [33,34], where they were found at increased concentrations in the post-mortem brain. Findings of PE alterations in experimental injury have been mixed: in moderate-severe TBI, PE levels are increased in lesioned brain tissue at three days [35]. Moreover, in moderate-severe TBI, decreased serum PE levels have been reported across the first seven days [25] and at three months, coinciding with decreases in brain tissue [36]. A recent study also described serum PE decreases at 24 h after single mTBI and repeated mTBI [37]. In contrast, in a mouse model of repeated mTBI, total PE concentrations significantly increased in the hippocampus at 24 h after injury, and remained significantly elevated to 12 months post-injury [38]. Given the roles for PE in myelin maintenance at the axonal node and vesicular formation [39,40], this generalised mTBI-mediated increase in PE species at 11 days post-injury may reflect heightened transport for reparative myelination at the node of Ranvier, for which we have previously described significant alteration after rmTBI in this model [19]. In contrast, several lipids were decreased only in response to 2× mTBI, including lysophosphatidylethanolamine (LPE) (20:1), phosphocholine (PC) (20:0/22:4), and phosphoinositol (PI) (18:1/18:2) and (20:0/18:2). LPEs have recently been shown to stimulate neurite outgrowth and reduce glutamate-mediated excitotoxicity in neuronal culture [41], indicating a beneficial role after injury. Likewise, PIs are key regulators of cytoskeletal function and myelination, with tissue depletion linked to demyelination and axonal loss [42], and PCs have a vital role in the structural integrity of the neuronal and glial membranes [43].

While future mechanistic studies are required to identify the damage processes responsible for the subacute lipid alterations reported in this study, we previously found increased lipid peroxidation in cortical neurons at day four following 2× mTBI [21], suggesting that oxidative stress-induced lipid peroxidation may play a role. Lipid peroxidation is a hallmark of neurodegenerative conditions, such as TBI [44]. The prolonged peroxidation of phospholipids causes cell membrane disintegration [45]. As PC and PE are the major lipids in neuronal somas and neuritic processes [46], their oxidative damage in the brain could potentially stimulate increased recruitment of PC, PE and their metabolites from the periphery for repairing neurons. The observed plasma decreases in LPE, PC and PI concentrations after 2× mTBI may therefore reflect a higher brain demand for reparative purposes. Further investigation of brain tissue lipid species will be required to investigate this possibility.

The findings of this study need to be considered in the context of several limitations. We cannot be sure of the extent to which the lipids detected in extracranial plasma samples reflect the true nature of lipid alteration within brain tissue, or whether they are altered by peripheral factors. Future research should compare lipid profiles in plasma to those in the brain parenchyma and/or cerebrospinal fluid to establish lipid similarities or differences after mTBI. This will also be particularly valuable for studies of temporal lipid changes. We chose 11 days post-mTBI for our exploratory lipid analyses, as this time point is known to have particular pathological relevance in this model. However, the characterisation of alterations at additional acute and chronic time points is a key future direction to delineate whether lipid alteration is indicative of injury or repair responses.

Our study also included small cohorts of rats in each group, and thus may have been underpowered to detect additional statistical differences between groups. Further work is needed to determine if these preliminary findings are replicable in a larger cohort, and to discover whether any key lipid changes were missed in this initial analysis.

Finally, we cannot speak to whether our findings in female rats are translatable to males. Studies in females are lacking in TBI research, with comparatively little being known about female biological responses to injury [47]. Since female sex hormones may be neuroprotective following TBI [48], future studies including male rats are required to identify sex-dependent changes in lipid profiles. However, it is noteworthy that our study identified differences in lipid species between uninjured controls and 1× or 2× mTBI, indicating that the neuroprotective effects of female hormones are not absolute, and allowing the detection of injury-induced changes in female rats.

## 4. Materials and Methods

### 4.1. Animals and Ethics

Adult female Piebald Viral Glaxo rats (160–200 g; 3 months of age) from the Animal Resource Centre (Murdoch, WA, Australia) were used for this study. Animals were housed in pairs under specific pathogen-free conditions and a 12:12 h light:dark cycle. Rats had free access to food and water. They were acclimatised to housing conditions for a minimum of one week prior to any procedures. All procedures involving rats were approved by the University of Western Australia Animal Ethics Committee (Approval Number RA/3/100/1699), in accordance with the Australian Code of Practice for the Care and Use of Animals for Scientific Purposes, issued by the National Health and Medical Research Council.

### 4.2. Closed Head Weight-Drop Nodel of Repeated Mild Traumatic Brain Injury

Twenty rats were randomly assigned to the sham (*n* = 6), 1× mTBI (*n* = 7) or 2× mTBI (*n* = 7) groups. Injuries were delivered at 24 h intervals, using a closed head, rotational acceleration weight-drop model of mTBI which has been described in detail previously [21]. Briefly, rats were anaesthetised with 4% isoflurane in 4 L/min oxygen and their heads were shaved before they were laid prone on a delicate task wiper (Kimwipes, Kimberly-Clark, Irving, TX, USA). The rat’s head was aligned to ensure that the 250 g weight released from a 1 m height would impact midline 2–3 mm anterior to the front of the ears, at the lambda suture line. Sham procedures were identical except for the weight drop (i.e., no injury). Animals that received 1× mTBI received a sham procedure when mTBI was not administered, ensuring equal anaesthesia for all animals to control for the effect of anaesthesia. All rats received analgesia (Carprofen, 4 mg/kg, s.c., Norbrook Laboratories, Tullamarine, VIC, Australia) immediately after the sham or mTBI procedures, before they were returned to their housing.

### 4.3. Plasma Collection for Lipidomics

At day 11 after the first sham or mTBI (day 1), rats were deeply anesthetised using pentobarbital sodium (160 mg/kg i.p., Virbac; supplied by Provet, Malaga, WA, Australia). Peripheral blood (2 mL) was collected by cardiac puncture using a 23 G needle in a syringe, transferred to an EDTA vacutainer (BD Vacutainer; REF 367839; BD Biosciences, North Ryde, NSW, Australia) and mixed by gentle inversion of the tubes. Plasma was separated from whole blood by centrifugation at 1500× *g* for 15 min at 4 °C, and stored at −80 °C until use.

### 4.4. Sample Preparation for Liquid Chromatography–Mass Spectrometry (LC-MS)

Lipidomic analysis was completed using an established method previously described [49], resulting in the quantification of 856 lipid species from 10 μL of rat plasma. Sample extraction was completed using a Biomek i5 sample automation system (Beckman Coulter, Mount Waverley, VIC Australia). To each 10 μL plasma sample, 90 μL of isopropanol (Fisher Scientific, Malaga, Western Australia) containing stable isotope-labelled internal standards (LipidyzerTM Internal Standards Kit from Sciex, Framingham, MA, USA, and SPLASH LipidoMIXTM, Lyso PI 17:1, Lyso PG 17:1, and Lyso PS 17:1, Avanti Polar Lipids, Alabaster, AL, USA) was added, and then mixed for 20 min. Samples were then centrifuged (3500× *g* for 10 min at 4 °C). Supernatant aliquots of 70 μL were then transferred into an Eppendorf 350 μL 96-well plate for LC-MS analysis. Samples were analysed within 24 h of preparation. QC samples were prepared using a commercial pooled human plasma sample (BioIVT, New York, NY, USA) and underwent replicate extraction (*n* = 5) using the method described. These replicate samples were periodically analysed throughout the analytical sequence.

### 4.5. Liquid Chromatography Mass Spectrometry (LC-MS)

Lipidomic analysis of the plasma sample extracts was performed using a targeted tandem mass spectrometry (MS/MS) approach using predefined multiple reaction monitoring (MRM) transitions (Sciex sMRM Pro Builder, Framingham, MA, USA) and in-house defined chromatographic retention time windows.

The analytical system consisted of a Sciex ExionLC ^TM^ coupled to a Triple Quadrupole Linear Ion Trap (QTRAP 6500+) (SCIEX, Concord, ON, Canada). Lipid metabolites were chromatographically separated using a Waters Acquity BEH C18 reverse phase column (1.7 µm, 100 × 2.1 mm particle size; Waters Corp., Milford, MA, USA), maintained at 60 °C.

The mobile phase consisted of water, acetonitrile and isopropanol (all Optima grade), which were all purchased from Thermo Fisher Scientific (Malaga, Western Australia). Mobile phase composition and gradients are described in detail in the Appendix A, along with additional instrument settings. Specific MS settings, MRM transitions and chromatographic retention times for the lipids of interest are reported in the results section and Appendix A are included in Appendix A.

### 4.6. Data Pre-Processing

Peak picking was conducted using SkylineMS [50] (Version 21.1) MacCoss Lab Software, University of Washington, Seattle, WA, USA). Data preprocessing and feature filtering was performed using R (version 4.4.1; R Foundation, Vienna, Austria) in R Studio [51] (version 1.4.1; R Studio, Boston, MA, USA) using in-house scripts; first, the lipid features were kept in the dataset if the relative standard deviation across the replicate QC samples was <30%; signal drift correction was then performed using the package statTarget [52]; the data then underwent logarithmic transformation to normalise variance and to estimate the normal distribution required for the univariate and multivariate analyses.

### 4.7. Statistical Analyses

For multivariate statistics, the final data matrix was transferred to SIMCA 17.0.1 (Sartorius AG, Goettingen, Germany). Data were pareto-scaled prior to multivariate modelling. PCA was performed to assess the measurement precision of the QC data, as well as to identify and remove the outlier samples. Three outlier samples (sham group = 2, 1× mTBI group = 1) were identified within the PCA via Hotelling’s T2 (95% CI) and removed from subsequent statistical analysis. Following PCA analysis, a supervised OPLS-DA was performed. Variable Importance for Projection Plots (VIP) were used to determine the importance of each lipid in the OPLS-DA model.

Following multivariate analysis, and to further stratify the three groups, univariate tests were performed using a Kruskal–Wallis one-way ANOVA on ranks, with Dunn’s test applied post-hoc to determine intergroup differences (sham, 1× mTBI and 2× mTBI). Statistical significance was deemed where *p* ≤ 0.05. As the study was of a discovery design and to avoid the potential loss of true positives, no correction for multiple testing was performed. Univariate analysis was completed using R (version 4.4.1) in R Studio [51] (version 1.4.1). Box plots were produced using GraphPad Prism v8.0.0 (GraphPad Software, San Diego, CA, USA, www.graphpad.com (accessed on 28 October 2021)).

## 5. Conclusions

Lipids are key molecules in a plethora of biochemical and physiological processes, including cellular signalling, immune responses, membrane lipid structure, and inflammatory pathways. Measuring the perturbations of the lipid species in clinically-relevant mTBI models provides further insight into the underlying pathophysiology, and highlights lipid alteration as a potential therapeutic target to improve outcomes following mTBI. The findings from our lipidomic analysis build on reported TBI insights in the literature. Furthermore, we provide novel insights into the subacute lipidomic signature of single and repeated mTBI. Given our finding of bi-directional plasma lipid concentration changes with single and repeated injury, the alteration of lipid species may be a new avenue for biomarker research in mTBI. Future work is needed to establish the degree to which pathological brain changes are reflected by plasma lipids, and to characterise the temporal lipid profile after single and repeated injury.

## Figures and Tables

**Figure 1 metabolites-12-00322-f001:**
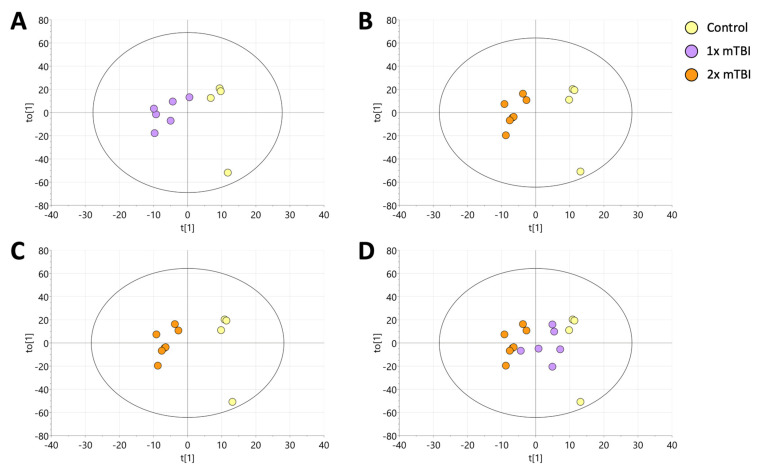
Orthogonal projection to latent structure-discriminant analysis (OPLS-DA) of plasma lipids following sham, 1× or 2× mTBI. (**A**) OPLS-DA visualisations of control samples vs. 1× mTBI samples. (**B**) OPLS-DA visualisations of 2× mTBI vs. 1× mTBI. The model was scaled proportionally to R2x, R2x(1) = 0.143, R2Xo(1) = 0.415. (**C**), OPLS-DA visualisations of control vs. 2× mTBI. (**D**) singular mTBI intervention (1× mTBI) was projected onto the final model for validation. The model was scaled proportionally to R2x, R2x(1) = 0.0886, R2Xo(1) = 0.548. mTBI, mild traumatic brain injury.

**Figure 2 metabolites-12-00322-f002:**
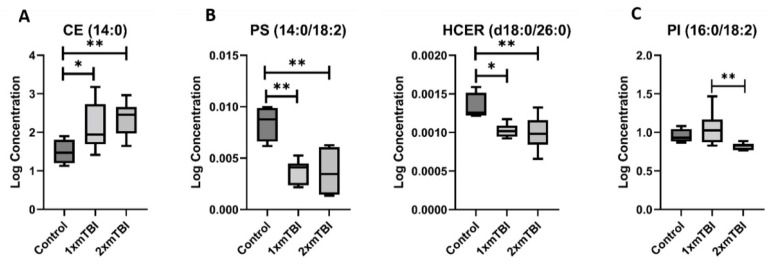
Lipids that change with one and two injuries vs. sham control. (**A**), increase after 1× or 2× mTBI; (**B**), decrease after 1× or 2× mTBI; (**C**), increase with 1× mTBI and decrease with 2× mTBI. Boxplots represent the minimum, first quartile, median, third quartile, and maximum, for the control, 1× mTBI and 2× mTBI groups. Significant differences between groups are indicated above the boxplots by * (*p* < 0.05) or ** (*p* < 0.01). CE, cholesterol ester; HCER, hexosylceramide; mTBI, mild traumatic brain injury; PI, phosphoinositol; PS, phosphoserine.

**Figure 3 metabolites-12-00322-f003:**
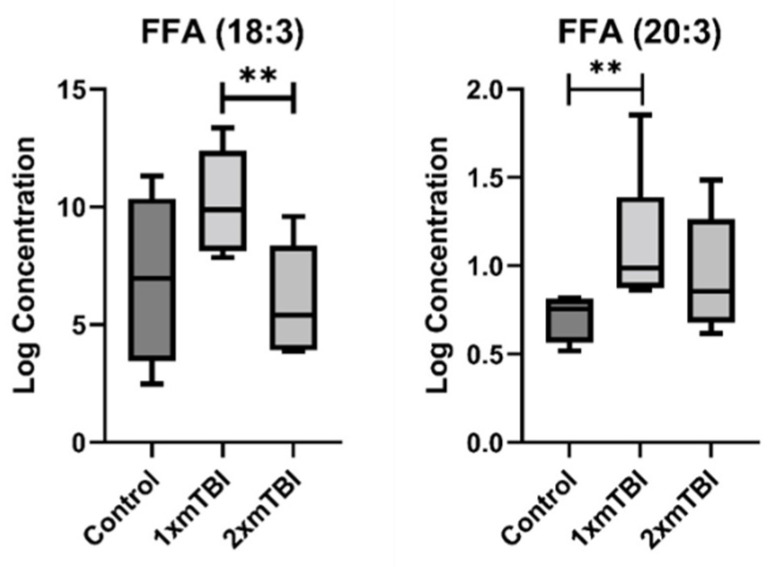
Lipids that change with one injury vs. sham control. Boxplots represent the minimum, first quartile, median, third quartile, and maximum, for the control, 1× mTBI and 2× mTBI groups. Significant differences between groups are indicated above the boxplots by ** (*p* < 0.05) or ** (*p* < 0.01). FFA, free fatty acid; mTBI, mild traumatic brain injury.

**Figure 4 metabolites-12-00322-f004:**
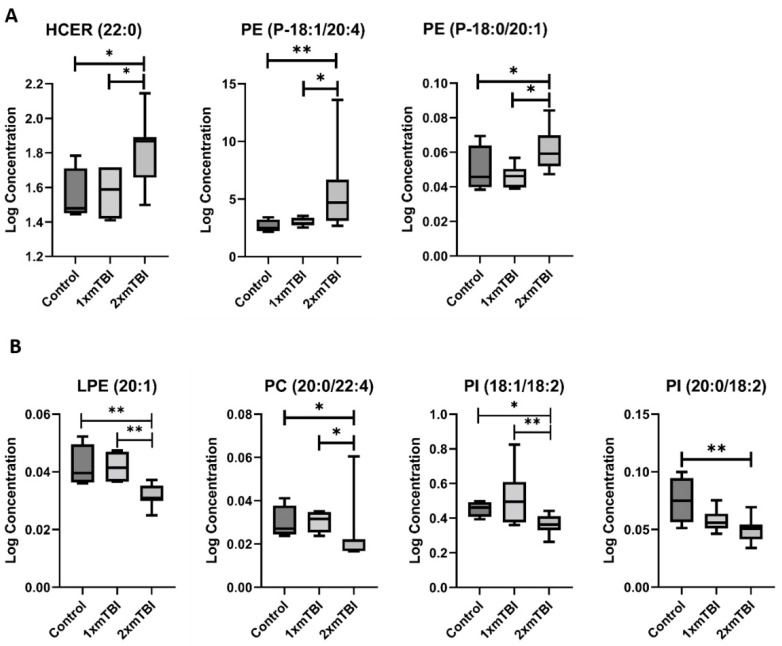
Lipids that change with two injuries vs. sham control. (**A**), Increase after 2× mTBI; (**B**), decrease after 2× mTBI. Boxplots represent the minimum, first quartile, median, third quartile, and maximum, for the control, 1× mTBI and 2× mTBI groups. Significant differences between groups are indicated above the boxplots by * (*p* < 0.05) or ** (*p* < 0.01). HCER, hexosylceramide; LPE, lysophosphatidylethanolamine; mTBI, mild traumatic brain injury; PC, phosphocholine; PE, phosphoethanolamine; PI, phosphoinositol.

**Table 1 metabolites-12-00322-t001:** Kruskal–Wallis one-way ANOVA on ranks analysis of plasma lipids.

Lipid Class	Lipid Species	Kruskal-Wallis		Post-Hoc Dunn’s Test			Control Mean (95% CI)	1× mTBI Mean (95% CI)	2× mTBI Mean (95% CI)
		*p*	eta^2^	*p* (Control vs. 1× mTBI)	*p* (Control vs. 2× mTBI)	*p* (1× mTBI vs. 2× mTBI)			
LPE	LPE(20:1)	0.008	0.554	0.459	0.008	0.002	0.042 (0.030,0.054)	0.042 (0.036,0.048)	0.032 (0.028,0.036)
LPE	LPE(22:6)	0.009	0.537	0.076	0.103	0.001	0.038 (0.022,0.054)	0.049 (0.039,0.059)	0.029 (0.023,0.035)
PI	PI(16:0/18:2)	0.014	0.471	0.449	0.013	0.004	0.952 (0.806,1.098)	1.050 (0.815,1.285)	0.819 (0.777,0.861)
LPE	LPE(22:5)	0.016	0.443	0.092	0.12	0.002	0.015 (0.007,0.023)	0.021 (0.013,0.029)	0.010 (0.006,0.014)
LPG	LPG(18:2)	0.016	0.443	0.13	0.084	0.002	0.016 (0.011,0.021)	0.018 (0.015,0.021)	0.012 (0.009,0.015)
PS	PS(14:0/18:2)	0.018	0.43	0.006	0.005	0.48	0.008 (0.005,0.011)	0.004 (0.003,0.005)	0.004 (0.002,0.006)
PE	PE(P-18:1/20:4)	0.025	0.384	0.166	0.005	0.036	2.640 (1.771,3.511)	2.990 (2.596,3.376)	5.630 (2.151,9.101)
PI	PI(16:0/18:3)	0.029	0.363	0.076	0.189	0.004	0.006 (0.005,0.007)	0.009 (0.006,0.012)	0.005 (0.004,0.006)
HCER	HCER(d18:0/26:0)	0.032	0.348	0.015	0.006	0.387	0.001 (0.001,0.001)	0.001 (0.001,0.001)	0.001 (0.001,0.001)
FFA	FFA(20:3)	0.038	0.324	0.005	0.057	0.118	0.712 (0.492,0.932)	1.140 (0.738,1.532)	0.954 (0.663,1.245)
PI	PI(18:1/18:2)	0.039	0.319	0.429	0.03	0.01	0.454 (0.383,0.525)	0.516 (0.341,0.691)	0.361 (0.306,0.416)
HCER	HCER(22:0)	0.04	0.317	0.469	0.021	0.013	1.550 (1.292,1.800)	1.570 (1.431,1.715)	1.820 (1.627,2.003)
CE	CE(14:0)	0.041	0.312	0.043	0.006	0.198	1.490 (0.988,2.000)	2.140 (1.475,2.805)	2.370 (1.967,2.775)
FFA	FFA(18:3)	0.049	0.288	0.054	0.322	0.008	6.930 (1.147,12.715)	10.200 (7.815,12.607)	5.940 (3.900,7.970)
PI	PI(20:0/18:2)	0.049	0.287	0.399	0.039	0.011	0.030 (0.018,0.042)	0.03 (0.025,0.035)	0.024 (0.009,0.039)
PC	PC(20:0/22:4)	0.049	0.287	0.399	0.039	0.011	0.050 (0.028,0.072)	0.046 (0.039,0.053)	0.061 (0.049,0.073)
PE	PE(P-18:0/20:1)	0.049	0.288	0.159	0.009	0.064	0.075 (0.043,0.107)	0.058 (0.048,0.068)	0.049 (0.039,0.059)
PI	PI(18:1/18:1)	0.05	0.286	0.237	0.081	0.008	0.498 (0.278,0.718)	0.610 (0.371,0.849)	0.385 (0.301,0.469)

Pairwise post-hoc Dunn’s tests were performed on lipids that returned a Kruskal–Wallis *p* < 0.05. Kruskal–Wallis effect size (eta^2^), individual group means and their 95% confidence intervals are also displayed. CE, cholesterol ester; FFA, free fatty acid; HCER, hexosylceramide; LPE, lysophosphatidylethanolamine; LPG, lysophosphoglycerol; mTBI, mild traumatic brain injury; PC, phosphocholine; PE, phosphoethanolamine; PI, phosphoinositol; PS, phosphoserine.

## Data Availability

The data that support the findings of this study are available on request from the corresponding author. The data are not publicly available due to privacy or ethical restrictions.

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
