# Peer review of "Plasma Lipid Profiles Change with Increasing Numbers of Mild Traumatic Brain Injuries in Rats"

_metabolites, 2022, doi:10.3390/metabo12040322_

Round 1

Reviewer 1 Report

The paper presented by Anyaegbu et al is a robust analytical hypothesis generation study into the changes in lipid populations within mTBI. The methods within the paper are sound and well established by the group. Data is analysed well and clearly, the methods cover all of the aspects needed. it appears that the ethical consideration to conduct the work is in place. The outcome is the possible identification of biomarkers and the results are of interest to the field. 

Minor Comments

Figure 1 has red and green dots that the colour blind can not tell apart can these be changed in some way.

Some further discussion (one paragraph or less) into the mechanism that results in the given lipids being altered would be a benefit to the reader. We are left with a list but some indication as to why these specific lipids would be appreciated. However, the mechanisms might not be known and if so this could be pointed out. 

A brief outline of how the in house scripts operate could be added to the Data pre-processing section.

Author Response

Minor Comments

Figure 1 has red and green dots that the colour blind can not tell apart can these be changed in some way. 

Author response: We thank the reviewer for this observation and have now updated the figure to a colourblind-friendly colour palette.

Some further discussion (one paragraph or less) into the mechanism that results in the given lipids being altered would be a benefit to the reader. We are left with a list but some indication as to why these specific lipids would be appreciated. However, the mechanisms might not be known and if so this could be pointed out.

Author response: We have now added an additional paragraph to the Discussion section, in which we have highlighted lipid peroxidation as a potential mechanism that could be responsible for the alterations in lipid species reported:

“While future mechanistic studies are required to identify damage processes responsible for the subacute lipid alterations reported in this study, we previously found increased lipid peroxidation in cortical neurons at day 4 following 2x mTBI [21] suggesting that oxidative‑stress induced lipid peroxidation may play a role. Lipid peroxidation is a hallmark of neurodegenerative conditions like TBI [49]. Prolonged peroxidation of phospholipids causes cell membrane disintegration [50]. As PC and PE are the major lipids in neuronal somas and neuritic processes [51], their oxidative damage in the brain could potentially stimulate increased recruitment of PC and PE and their metabolites from the periphery for repairing neurons.”

A brief outline of how the in-house scripts operate could be added to the Data pre-processing section. 

Author response: We have updated the data pre-processing section to include more detail. It now reads as follows:

4.6.       Data pre processing 

Peak picking was conducted using SkylineMS [24] (Version 21.1). Data preprocessing and feature filtering was performed using R (version 4.4.1) in R Studio [25] (version 1.4.1) using in-house scripts; first lipid features were kept in the dataset if the relative standard deviation across the replicate QC samples was < 30%; Signal drift correction was then performed using the package statTarget [26]; data then underwent logarithmic transformation to normalise variance and to estimate the normal distribution required for univariate and multivariate analysis. 

Reviewer 2 Report

General comments

The manuscript is generally well written and focuses on an interesting topic. There is compelling evidence in support of the importance of this problem and the proposed solution to study mild traumatic brain injuries in rats. This study is building on extensive rat models of head injury research by this team.

Specific Comments by Section

Introduction

A lot of unfamiliar terminology is used as well as subsequent abbreviations which tended to make it difficult to follow. Perhaps adding an abbreviations section could help readers not that familiar with the topic to follow along.

It seemed that a core set of lipids - PC, LPC, PE, LPE – could be defined as such and then additional ones included to better illustrate what is common in the previous literature and what unique lipids were identified. Or add a table in the supplementary materials. This would also help place the results of this work in context with previous research to identify the unique and confirmatory contributions.

The rational for studying rotational movements was not clear – how is this defined and how would this alter the lipid profiles found in rats or other models?

I presumed sham was the control condition that did not involve any head trauma but it would be important to define this more clearly somewhere.

Methods

The pre-processing of the data and use of O-PLS-DA seemed appropriate. However, although the VIP values were calculated, they did not seem to used to identify lipids important to the O-PLS-DA models. Why not?

Why were the regression coefficient plots not used to also identify important lipids, especially those with high abundance levels and narrow jackknife confidence intervals?

Why was Kruskal- Wallis (K-W) analysis undertaken when the VIP scores could be used in place of this additional evaluation? No adjustment for multiple testing was made in the K-W ANOVAs which is a substantial issue when 856 lipids are evaluated one at a time.

There are no details on how the AUC values were calculated given the extremely small groups sizes. Usual k-fold cross validation would not work in this setting and there was no mention of a test validation set. If the same data set used to create the model and then used to predict from it, that is not correct. This may explain the extremely high AUC values that appear suspicious especially when compared with the Q2 values.

Were any lipids evaluated as a priori hypotheses? Based on the previous findings noted in the introduction, some lipids could have been tested specifically for their presence or absence.

Was a pathway analysis considered?

Results

The results are based on extremely small sample sizes in each group and the results need to be interpreted in this context as very preliminary. More discussion about the limitations of these findings need to be included in the discussion.

The AUC values do not appear to be correctly calculated, as noted above. If some sort of k-fold cross validation was used, this needs to be mentioned. Otherwise, these values should be removed.

The difference between the R2Y values and Q2Y suggests very substantial overfitting in your models. The Q2Y values are low but not unexpected for studies in animal models. These values suggest extreme caution in the utility of your findings as they are not that predictive even in your own data. 

The model that was fit in Figure 1D was not entirely clear to me in terms of what was done; it would benefit from more description.

Discussion/Conclusions

Limited discussion of future research directions, such as confirming these findings, evaluating the longevity of these effects and determining injury response versus repair response to better characterize their findings.

Really no mention of limitations of this study, which need to be included, as the small sample sizes and no validation data set will reduce the strength of the conclusions.

Author Response

Specific Comments by Section

Introduction

A lot of unfamiliar terminology is used as well as subsequent abbreviations which tended to make it difficult to follow. Perhaps adding an abbreviations section could help readers not that familiar with the topic to follow along. 

Author response: A list of abbreviations used in the manuscript has been provided to aid the reader as suggested. We have also removed all non-essential abbreviations.

It seemed that a core set of lipids - PC, LPC, PE, LPE – could be defined as such and then additional ones included to better illustrate what is common in the previous literature and what unique lipids were identified. Or add a table in the supplementary materials. This would also help place the results of this work in context with previous research to identify the unique and confirmatory contributions. 

Author response: We thank the reviewer for this suggestion. We have now included a supplementary table (Table S1) of findings describing lipid changes in commonly examined classes in TBI, and included our findings in this table to give at-a-glance context to our findings within the literature, referring to the Supplementary Table in the Discussion.

The rational for studying rotational movements was not clear – how is this defined and how would this alter the lipid profiles found in rats or other models? 

Author response: We thank the reviewer for this observation. We have now clarified the definition and rationale for studying rotational movements in the Introduction (Lines 96-99).

I presumed sham was the control condition that did not involve any head trauma but it would be important to define this more clearly somewhere.

Author response: We have now clarified that the Sham group of animals did not receive injury, in the Abstract (Line 24) and Methods (Lines 326-327).

Methods

The pre-processing of the data and use of O-PLS-DA seemed appropriate. However, although the VIP values were calculated, they did not seem to used to identify lipids important to the O-PLS-DA models. Why not?

Author response: We thank the reviewer for this observation, VIP values that most influenced each of the OPLS-DA models have now been included in the supplementary file for the reader’s reference, via a ranked list. The intention for the inclusion of the OPLS-DA models in the study was to provide a visualisation for the reader that multivariate differences existed in the plasma lipidome when performing two-way group vs group comparisons (e.g. control vs 1x mTBI; control vs 2x mTBI and 1x mTBI vs 2x mTBI). This was then followed by three-way univariate analysis (Kruskal-Wallis, and post-hoc Dunn’s test) for identification of key lipid differences existing across the three groups.

Why were regression coefficient plots not used to also identify important lipids, especially those with high abundance levels and narrow jackknife confidence intervals?

Author response: We thank the reviewer for the suggestion, however in this study we elected to employ a combination of multivariate OPLS-DA modelling as a tool to visualise lipid differences between groups and univariate Kruskal-Wallis testing to report key lipids that differ across the three groups. However, we have now included 95% confidence intervals for the individual group means and Kruskal-Wallis effect size (eta2) in results Table 1 to provide the reader with additional information on the data analysis.

Why was Kruskal- Wallis (K-W) analysis undertaken when the VIP scores could be used in place of this additional evaluation?

Author response: The OPLS-DA models were performed on a group vs group basis primarily as a visualisation tool to demonstrate multivariate differences between groups, however on the advice of the reviewer, VIP scores for each of the models have now included in supplementary file; control vs 1xTBI; control vs 2xTBI; and 1xTBI vs 2xTBI.

In addition to the OPLS-DA modelling, we also included the additional Kruskal-Wallis univariate testing with post-hoc pairwise Dunn’s tests to demonstrate changes in individual lipids across all three of the groups, and to report lipids that changed between one inter-group comparison, but perhaps not a second.

No adjustment for multiple testing was made in the K-W ANOVAs which is a substantial issue when 856 lipids are evaluated one at a time.

Author response: As the study design was discovery based, was limited by the sample n and contained a high number of variables we elected not to correct for multiple for testing in this instance, to avoid being over conservative and missed reporting of biomarkers of interest. This has now been clarified in the methods.

There are no details on how the AUC values were calculated given the extremely small groups sizes. Usual k-fold cross validation would not work in this setting and there was no mention of a test validation set. If the same data set used to create the model and then used to predict from it, that is not correct. This may explain the extremely high AUC values that appear suspicious especially when compared with the Q2 values. 

Author response: We thank the reviewer for their comments, as the reviewer indicates k-fold cross validation was not possible with the number of samples in the study and the included ROC AUC values have not undergone a cross validation. With this in mind, we have removed the ROC analysis from the manuscript and instead have focused on the R2 and Q2 values provided from the OPLS-DA models. The results section now reads:

“To identify the metabolic signature associated with mTBI, supervised OPLS-DA was performed on the lipid data generated from sham (n = 4), 1xmTBI (n= 6) and 2x mTBI (n= 7) groups (Figure 1). The OPLS-DA model scores were as follows: sham control and 1x mTBI groups (Figure 1A) (R2 X = 0.637, R2 Y = 0.858, Q2 = 0.307); 1x mTBI and 2x mTBI groups (Figure 1B) (R2 X = 0.558, R2 Y = 0.84, Q2 = 0.206); sham control and 2x mTBI groups (Figure 1C) (R2 X = 0.603, R2 Y = 0.952, Q2 = 0.322).”

Were any lipids evaluated as a priori hypotheses? Based on the previous findings noted in the introduction, some lipids could have been tested specifically for their presence or absence. 

Author response: This study was set-up and designed as a discovery-based study, with no a priori hypotheses utilised in the study design. The primary aim of the study was to test to see if traumatic brain injury in rats also resulted in a wider systemic change in plasma lipids. Therefore we employed an unbiased discovery approach.  Future follow-up validation studies will look to validate the findings reported in the manuscript and compare to existing literature in the field.

Was a pathway analysis considered? 

Author response: We thank the reviewer for their suggestion, however in this instance a pathway analysis was not considered, as the primary aim of the manuscript was to test the hypothesis that that traumatic brain injury in rats also results in a wider systemic change in plasma lipids. This was demonstrated using a combination of multivariate (OPLS-DA) and univariate (Kruskal-Wallis) methods. Future validation work will seek to interrogate specific pathways of interest and their systemic biological implications.

Results

The results are based on extremely small sample sizes in each group and the results need to be interpreted in this context as very preliminary. More discussion about the limitations of these findings need to be included in the discussion. 

Author response: We have now added limitations to the discussion section; see response to this comment under Discussion/Conclusions for description of limitations and future directions.

The AUC values do not appear to be correctly calculated, as noted above. If some sort of k-fold cross validation was used, this needs to be mentioned. Otherwise, these values should be removed. 

Author response: We thank the reviewer for this observation, and on their advice we have now removed the AUC values from the manuscript and instead have focused on the R2 and Q2 values provided from the OPLS-DA models.

The difference between the R2Y values and Q2Y suggests very substantial overfitting in your models. The Q2Y values are low but not unexpected for studies in animal models. These values suggest extreme caution in the utility of your findings as they are not that predictive even in your own data.

Author response: We thank the reviewer for raising this comment. We agree that OPLS-DA models are prone to overfitting, particularly when using a small sample n and large number of variables. However, in our experience of applying metabolic phenotyping and subsequent OPLS-DA multivariate modelling to biological samples, values such as those reported are typical. As the reviewer states in their comment “The Q2Y values are low but not unexpected for studies in animal models”. Furthermore, in the context of this study the OPLS-DA models were primarily used as a visualisation tool to demonstrate multivariate differences which existed between the groups, indicating lipid perturbations in blood plasma following mTBI. Univariate three-way Kruskal-Wallis tests were then employed to identify lipids of significance across the three groups.

The model that was fit in Figure 1D was not entirely clear to me in terms of what was done; it would benefit from more description. 

Author response: We thank the reviewer for raising the issue that Figure 1D is not adequately described. We have now re-worded the section, and hope that it now reads better:

“Finally, to test for the existence of a progressive trend in the data following 1x mTBI and 2x mTBI interventions, the 1x mTBI data was predictively projected onto the existing OPLS-DA model for sham control vs 2xTBI. This resulted in the 1xTBI data projecting into the model space between the sham control and 2xTBI groups, indicating a progressive change in the lipid data: sham control à 1xTBI à 2xTBI (Figure 1D).”

Discussion/Conclusions

Limited discussion of future research directions, such as confirming these findings, evaluating the longevity of these effects and determining injury response versus repair response to better characterize their findings.

Author response: An extended section has now been added to the Discussion to include limitations and future directions (Lines 285-306). In this section we combine limitations and future directions, discussing:

  • The need for validation in brain parenchymal samples to clarify plasma reflection of brain lipid alteration;
  • Our use of a single sub-acute timepoint, and the need for studies profiling lipids in acute and chronic injury phases to determine injury vs. repair responses;
  • The small cohort of rats used in this study and the need for replication in larger samples;
  • The use of female rats and the need for future comparison to male rats to determine whether there may be sex-specific differences.

Really no mention of limitations of this study, which need to be included, as the small sample sizes and no validation data set will reduce the strength of the conclusions.

Author response: This comment has been addressed in the response to the comment above.

Reviewer 3 Report

Comments for authors
This paper entitled “Plasma lipid profiles change with increasing numbers of mild traumatic brain injuries in rats” by Anyaegbu et al. reported lipid profiles of the rat plasma from different groups including sham, one (1x) or two (2x) mTBI using LC-MS. In addition, eighteen lipid species were identified to distinguish sham, 1x, and 2x mTBI through the multivariate OPLS-DA. Authors claimed that the alternation of lipids induced by increasing numbers of mTBI likely reflects a balance of damage and reparative responses. However, in my opinion, the high-resolution MS and MS/MS evidence should be added to validate the identification of 18 differentially expressed lipid species.

Some minor points.
1.    In the whole manuscript, the annotation of lipids should be consistent. There are several different names confusion. Examples of these confusion include but are not limited to PS(14:0,18:2) or PS(14:0/18:2) or PS(14:0_18:2); HCER(d18:0/26) or HCER(d18:0/26:0) or HCER(d18:0_26:0); FFA18:3 or FFA(18:3). Authors should check and revise them carefully. 
2.    In the Keywords, “mTBI” and the abbreviation “(LC-MS)” could be deleted. 
3.    The abbreviation “(SM)” should be added after sphingomyelin in Line 68 instead of in Line 84.

Author Response

In my opinion, the high-resolution MS and MS/MS evidence should be added to validate the identification of 18 differentially expressed lipid species.

Author response: We thank the reviewer for their comment, however in this instance the lipidomics data was generated using a SCIEX Exion Triple Quadrupole Linear Ion Trap (QTRAP) 6500+ MS Platform, and not on a high resolution mass spectrometry instrument. Lipid annotations were defined using in silico generated MS/MS transitions and retention time windows specifically set to capture the target lipids and avoid incorrect annotation. This has now been clarified in the methods section:

“Lipidomic analysis of the plasma sample extracts was performed using a targeted tandem mass spectrometry (MS/MS) approach using predefined multiple reaction monitoring (MRM) transitions (Sciex sMRM Pro Builder, Framingham, MA, USA) and in-house defined chromatographic retention time windows.

The analytical system consisted of a Sciex ExionLC TM coupled to a Triple Quadrupole Linear Ion Trap (QTRAP 6500+) (SCIEX, Concord, ON, Canada).  Lipid metabolites were chromatographically separated using a Waters Acquity BEH C18 reverse phase column (1.7 µm, 100 x 2.1 mm particle size; Waters Corp., Milford, MA, USA), maintained at 60°C. Full instrument settings are described in the Supplementary methods.”

Some minor points.

  1. In the whole manuscript, the annotation of lipids should be consistent. There are several different names confusion. Examples of these confusion include but are not limited to PS(14:0,18:2) or PS(14:0/18:2) or PS(14:0_18:2); HCER(d18:0/26) or HCER(d18:0/26:0) or HCER(d18:0_26:0); FFA18:3 or FFA(18:3). Authors should check and revise them carefully. 

Author response: We have carefully reviewed the manuscript and corrected these typographical errors in text and in figures.

  1. In the Keywords, “mTBI” and the abbreviation “(LC-MS)” could be deleted. 

Author response: These have now been removed from the Keywords as suggested.

  1. The abbreviation “(SM)” should be added after sphingomyelin in Line 68 instead of in Line 84. 

 Author response: The abbreviation “SM” has now been placed after first mention of sphingomyelin.

Reviewer 4 Report

  1. The English need improvement since there are some grammatical and syntax errors in the manuscript. For example,
  • in line number 43, the word “increased” may be as “an increased”;
  • in line number 52, “cell” as “the cell”;
  • in line number 112, “reproducibility” as “the reproducibility”;
  • in line number 326, “were used” as “was used”;
  • in line number 327, “Area” as “The area”;
  • in line number 328, “were used” as “was used”.

The grammar mistakes which are not mentioned here are also to be checked and corrected properly.

  1. There are some typing mistakes as well, and authors are advised to carefully proof-read the text. For example,
  • in line number 21, the words “blood brain” may be as “blood-brain”;
  • in line number 62, “lipid rich” as “lipid-rich”;
  • in line number 64, “acid containing” as “acid-containing”;
  • in line number 74, “membrane bound” as “membrane-bound”;
  • in line number 92, “month” as “months”;
  • in line number 95, “sub acute” as “subacute”;
  • in line number 99, “well characterised” as “well-characterised”;
  • in table 1, “p value” as “p-value”;
  • in line number 322, “outlier” as “outliers”..

The typos not mentioned here are also to be checked and corrected properly.

  1. Check the abbreviations throughout the manuscript and introduce the abbreviation when the full word appears the first time in the text and then use only the abbreviation (For example, central nervous system (CNS) and the authors have used “LPE” for both “lysophosphoethanolamine and lysophosphatidylethanolamine (is right form), similarly for “PS” “phosphoserine and phosphoinositols”, etc.,). And it should be in both abstract as well as in the remaining part of the manuscript. Make a word abbreviated in the article that is repeated at least three times in the text, not all words need to be abbreviated.

  1. The tables and figures legends should be improved and a proper footnote should be given. All legends should have enough description for a reader to understand tables and figures without having to refer back o the main text of the manuscript. For example, the necessary expansion may be given for abbreviations used.

  1. In the materials and methods, the author should include the source of chemicals used in this study.
  2. The authors have tried the animal model with females. Justification may be included why female rats are chosen for this study since in most of the animal studies usually male will be preferred due to hormonal influence in females.
  3. In conclusion, it is highly recommended to include weaknesses or limitations of the study and potential future research goals.

8. The reference cited in the conclusion section should be removed and it may be given in any other part of the manuscript. The conclusion should be key points of the overall observation of the present study only and not with others.

Author Response

Comments and Suggestions for Authors

The English need improvement since there are some grammatical and syntax errors in the manuscript. For example,

in line number 43, the word “increased” may be as “an increased”

in line number 52, “cell” as “the cell”;

in line number 112, “reproducibility” as “the reproducibility”;

in line number 326, “were used” as “was used”;

in line number 327, “Area” as “The area”;

in line number 328, “were used” as “was used”.

The grammar mistakes which are not mentioned here are also to be checked and corrected properly. 

There are some typing mistakes as well, and authors are advised to carefully proof-read the text. For example,

in line number 21, the words “blood brain” may be as “blood-brain”;

in line number 62, “lipid rich” as “lipid-rich”;

in line number 64, “acid containing” as “acid-containing”;

in line number 74, “membrane bound” as “membrane-bound”;

in line number 92, “month” as “months”;

in line number 95, “sub acute” as “subacute”;

in line number 99, “well characterised” as “well-characterised”;

in table 1, “p value” as “p-value”;

in line number 322, “outlier” as “outliers”

The typos not mentioned here are also to be checked and corrected properly.

Author response: We thank the reviewer for their careful review of our manuscript and their suggestions. We have thoroughly reviewed the manuscript for grammatical, syntactical and typographical errors. We have updated the manuscript to include compound adjectives and correct grammatical and typographical errors. In some instances we respectfully disagree with the reviewer and have not made the suggested change, e.g. previous line 322 - now lines 395 and 396 - where the text states “outlier samples” so a change to “outliers samples” would not be beneficial.

Check the abbreviations throughout the manuscript and introduce the abbreviation when the full word appears the first time in the text and then use only the abbreviation (For example, central nervous system (CNS) and the authors have used “LPE” for both “lysophosphoethanolamine and lysophosphatidylethanolamine (is right form), similarly for “PS” “phosphoserine and phosphoinositols”, etc.,). And it should be in both abstract as well as in the remaining part of the manuscript. Make a word abbreviated in the article that is repeated at least three times in the text, not all words need to be abbreviated. 

Author response: We thank the reviewer for highlighting these oversights and have now thoroughly reviewed all abbreviations and made corrections throughout the manuscript to ensure the correct abbreviation is used, and used after the first instance in text. We have intentionally included the full word as well as the abbreviation in the Discussion section, to aid readers who may be less familiar with lipids and their abbreviations. We have also included an abbreviation list for this purpose.

The tables and figures legends should be improved and a proper footnote should be given. All legends should have enough description for a reader to understand tables and figures without having to refer back to the main text of the manuscript. For example, the necessary expansion may be given for abbreviations used.

Author response: Figure and Table legends have now been revised for clarity, and improved as suggested to include all abbreviations and aid readability.

In the materials and methods, the author should include the source of chemicals used in this study. 

Author response: We thank the reviewer for this observation and have now updated the methods accordingly.

The authors have tried the animal model with females. Justification may be included why female rats are chosen for this study since in most of the animal studies usually male will be preferred due to hormonal influence in females. 

Author response: We thank the reviewer for this observation. We have added justification for using female rats in the Introduction (Line 107-109). We have also included a new section discussing the use of female rats with regard to hormonal influence and under-representation of females in the TBI literature (Lines 299-306). In this section we also highlight the need for future experiments to replicate our findings in male rats.

In conclusion, it is highly recommended to include weaknesses or limitations of the study and potential future research goals.

Author response: An extended section has now been added to the Discussion to include limitations and future directions (Lines 285-306). In this section we combine limitations and future directions, discussing:

  • The need for validation in brain parenchymal samples to clarify plasma reflection of brain lipid alteration;
  • Our use of a single sub-acute timepoint, and the need for studies profiling lipids in acute and chronic injury phases to determine injury vs. repair responses;
  • The small cohort of rats used in this study and the need for replication in larger samples;
  • The use of female rats and the need for future comparison to male rats to determine whether there may be sex-specific differences.

The reference cited in the conclusion section should be removed and it may be given in any other part of the manuscript. The conclusion should be key points of the overall observation of the present study only and not with others. 

Author response: Thank you to the reviewer for picking up on this oversight. We have now removed these references from the conclusion section.

Round 2

Reviewer 2 Report

Please define the column headers in Tables S1-S3.

Otherwise, the revised manuscript is more clearly written, the conclusions supported by the results and the limitations noted. The addition of the supplemental materials also adds value to understanding the study and its results. 

Author Response

Comments and Suggestions for Authors
Please define the column headers in Tables S1-S3.

Author response: We have now improved the Table captions to include improved definitions of the column headers.

Reviewer 3 Report

The revised manuscript by Anyaegbu et al. did not address my major concern yet. I regret to reject its publication on Metabolites at present form. In my opinion, the lipidomics strategy used in this work was not new, however, its application is very interesting and useful for the investigation of the blood lipids that may be related to mild traumatic brain injury. And the potential biomarkers indicated in this work will be very valuable for future study on the neuropathological state. Thus, the reliable identification of those lipid candidates is crucial to evaluate this work. I can not provide my comments on the results only based on the candidates' list shown in the present manuscript. Of cause, I am clear that Triple Quadrupole Linear Ion Trap (QTRAP) 6500+ MS is not a high-resolution MS platform, but the authors have to provide enough evidence to support the identification of those 18 lipid candidates. Even I could not find the information about how to identify lipids in Ref. 23, e.g., predefined MRM transitions.

Author Response

Comments and Suggestions for Authors

The revised manuscript by Anyaegbu et al. did not address my major concern yet. I regret to reject its publication on Metabolites at present form. In my opinion, the lipidomics strategy used in this work was not new, however, its application is very interesting and useful for the investigation of the blood lipids that may be related to mild traumatic brain injury. And the potential biomarkers indicated in this work will be very valuable for future study on the neuropathological state. Thus, the reliable identification of those lipid candidates is crucial to evaluate this work. I can not provide my comments on the results only based on the candidates' list shown in the present manuscript. Of cause, I am clear that Triple Quadrupole Linear Ion Trap (QTRAP) 6500+ MS is not a high-resolution MS platform, but the authors have to provide enough evidence to support the identification of those 18 lipid candidates. Even I could not find the information about how to identify lipids in Ref. 23, e.g., predefined MRM transitions.

Author response: We apologise to the reviewer for not adequately detailing the experiment in our previous response. We have now provided additional details in the form of Supplementary Table 5, which describes the specific mass spectrometry settings (Collision Energy (CE), Declustering Potential (DP), Entrance Potential (EP), Collision Cell Exit Potential (CXP), Internal Standard used (ISTD), the MRM transition details (Q1 Mass, Q3 Mass, adduct type) and retention times for the lipids of interest reported throughout the manuscript.

Round 3

Reviewer 3 Report

The revised manuscript partially address my major concern.